# Structural generalization in COGS: Supertagging is (almost) all you need

**Alban Petit**[1]  and  **Caio Corro**[2]  and  **François Yvon**[2]

[1]Université Paris-Saclay, CNRS, LISN, 91400, Orsay, France
[2]Sorbonne Université, CNRS, ISIR , F-75005 Paris, France

alban.petit@lisn.upsaclay.fr   {caio.corro,francois.yvon}@isir.upmc.fr

## Abstract

In many Natural Language Processing applications, neural networks have been found to fail to generalize on out-of-distribution examples. In particular, several recent semantic parsing datasets have put forward important limitations of neural networks in cases where compositional generalization is required. In this work, we extend a neural graph-based semantic parsing framework in several ways to alleviate this issue. Notably, we propose: (1) the introduction of a supertagging step with valency constraints, expressed as an integer linear program; (2) a reduction of the graph prediction problem to the maximum matching problem; (3) the design of an incremental early-stopping training strategy to prevent overfitting. Experimentally, our approach significantly improves results on examples that require structural generalization in the COGS dataset, a known challenging benchmark for compositional generalization. Overall, our results confirm that structural constraints are important for generalization in semantic parsing.

## 1 Introduction

Semantic parsing aims to transform a natural language utterance into a structured representation. However, models based on neural networks have been shown to struggle on out-of-distribution utterances where compositional generalization is required, *i.e.*, on sentences with novel combinations of elements observed separately during training (Lake and Baroni, 2018; Finegan-Dollak et al., 2018; Keysers et al., 2020). Jambor and Bahdanau (2022) showed that neural graph-based semantic parsers are more robust to compositional generalization than sequence-to-sequence (seq2seq) models. Moreover, Herzig and Berant (2021), Weißenhorn et al. (2022) and Petit and Corro (2023) have shown that introducing valency and type constraints in a structured decoder improves compositional generalization capabilities.

In this work, we explore a different method for compositional generalization, based on supertagging. We demonstrate that local predictions (with global consistency constraints) are sufficient for compositional generalization. Contrary to Herzig and Berant (2021) and Petit and Corro (2023), our approach can predict any semantic graph (including ones with reentrancies), and contrary to Weißenhorn et al. (2022) it does not require any intermediate representation of the semantic structure.

Moreover, our experiments highlight two fundamental features that are important to tackle compositional generalization in this setting. First, as is well known in the syntactic parsing literature, introducing a supertagging step in a parser may lead to infeasible solutions. We therefore propose an integer linear programming formulation of supertagging that ensures the existence of at least one feasible parse in the search space, via the so-called companionship principle (Bonfante et al., 2009, 2014). Second, as the development dataset used to control training is in-distribution (*i.e.*, it does not test for compositional generalization), there is a strong risk of overfitting. To this end, we propose an incremental early-stopping strategy that freezes part of the neural network during training.

Our contributions can be summarized as follows:

- we propose to introduce a supertagging step in a graph-based semantic parser;

- we show that, in this setting, argument identification can be reduced to a matching problem;

- we propose a novel approach based on inference in a factor graph to compute the weakly-supervised loss (*i.e.*, without gold alignment);

- we propose an incremental early-stopping strategy to prevent overfitting;

- we evaluate our approach on COGS and observe that it outperforms comparable baselines on compositional generalization tasks.

| | Training example | Generalization example |
|---|---|---|
| **Lexical generalizations** | | |
| Subj to obj (common) | A **hedgehog** ate the cake | The baby liked the **hedgehog** |
| Prim to subj (proper) | **Paula** | **Paula** sketched William |
| Active to passive | The crocodile **blessed** William | A muffin **was blessed** |
| PP dative to double dative | Jane **shipped** the cake **to** John | Jane **shipped** John the cake |
| Agent NP to unaccusative | The cobra **helped** a dog | The cobra **froze** |
| **Structural generalizations** | | |
| Obj to subj PP | Noah ate **the cake on the plate** | **The cake on the table** burned |
| PP recursion | Ava saw the ball **in the bottle** | Ava saw the ball **in the bottle on the table on the floor** |
| CP recursion | Emma said **that** the cat danced | Emma said **that** Noah knew **that** Lucas saw **that** the cat danced |

Table 1: Examples of lexical generalization and structural generalization from COGS, adapted from (Kim and Linzen, 2020, Table 1). For PP and CP recursions, the number of recursions observed at test time is greater than the number of recursions observed during training.

**Notations.** A set is written as $\{\cdot\}$ and a multiset as $[\![\cdot]\!]$. We denote by $[n]$ the set of integers $\{1, ..., n\}$. We denote the sum of entries of the Hadamard product as $\langle \cdot, \cdot \rangle$ (*i.e.*, the standard scalar product if arguments are vectors). We assume the input sentence contains $n$ words. We use the term "concept" to refer to both predicates and entities.

## 2 Semantic parsing

### 2.1 COGS

The principle of compositionality states that

> "The meaning of an expression is a function of the meanings of its parts and of the way they are syntactically combined." (Partee, 1984)

Linguistic competence requires compositional generalization, that is the ability to understand new utterances made of known parts, *e.g.*, understanding the meaning of "Marie sees Pierre" should entail the understanding of "Pierre sees Marie".

The *Compositional Generalization Challenge based on Semantic Interpretation* (COGS, Kim and Linzen, 2020) dataset is designed to evaluate two types of compositional generalizations. First, *lexical generalization* tests a model on known grammatical structures where words are used in unseen roles. For example, during training the word hedgehog is only used as a subject; the model needs to generalize to cases where it appears as an object. Second, *structural generalization* tests a model on syntactic structures that were not observed during training. Illustrations are in Table 1.

The error analysis presented by Weißenhorn et al. (2022) emphasizes that neural semantic parsers

achieve good accuracy for lexical generalization but fail for structural generalization.

**Semantic graph construction.** A semantic structure in the COGS dataset is represented as a logical form. We transform this representation into a graph as follows:

1. For each concept instance, we add a labeled vertex.

2. For each argument $p$ of a concept instance $p'$, we create a labeled arc from the vertex representing $p'$ to the vertex representing $p$.

3. COGS explicitly identifies definiteness of nouns. Therefore, for each definite noun that triggers a concept, we create a vertex $p$ with label `definite` and we create an arc with label `det` from $p$ to the vertex representing the noun's concept.[1] Indefiniteness is marked by the absence of such structure.

This transformation is illustrated in Figure 1.

### 2.2 Graph-based decoding

The standard approach (Flanigan et al., 2014; Dozat and Manning, 2018; Jambor and Bahdanau, 2022) to graph-based semantic parsing is a two-step pipeline:

1. concept tagging;

2. argument identification.

---

[1] Using the determiner as the head of a relation may be surprising for readers familiar with syntactic dependency parsing datasets, but there is no consensus among linguists about the appropriate dependency direction, see *e.g.*, Müller (2016, Section 1.5) for a discussion.

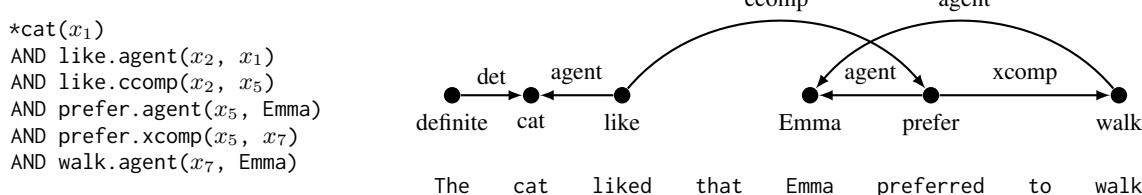

Figure 1: **(left)** Semantic analysis of the sentence "The cat liked that Emma preferred to walk" in the COGS formalism. A * denotes definiteness of the following predicate. **(right)** Graph-based representation of the semantic structure. Note that we mark definiteness using an extra vertex anchored on the determiner.

The second step is a sub-graph prediction problem.

**Concept tagging.** We assume that each word can trigger at most one concept. Let $T$ be the set of concepts, including a special tag $\emptyset \in T$ that will be used to identify semantically empty words (*i.e.*, words that do not trigger any concept). Let $\boldsymbol{\lambda} \in \mathbb{R}^{n \times T}$ be tag weights computed by the neural network. Without loss of generality, we assume that $\lambda_{i,\emptyset} = 0, \forall i \in [n]$. We denote a sequence of tags as a boolean vector $\boldsymbol{x} \in \{0,1\}^{n \times T}$ where $x_{i,t} = 1, t \neq \emptyset$, indicates that word $i$ triggers concept $t$. This means that $\forall i \in [n], \sum_{t \in T} x_{i,t} = 1$. Given weights $\boldsymbol{\lambda}$, computing the sequence of tags of maximum linear weight is a simple problem.

**Argument identification.** We denote $L$ the set of argument labels, *e.g.*, agent $\in L$. The second step assigns arguments to concepts instances. For this, we create a labeled graph $G = (V, A)$ where:

- $V = \{i \in [n] | x_{i,\emptyset} \neq 0\}$ is the set of vertices representing concept instances;

- $A \subseteq V \times V \times L$ is the set of labeled arcs, where $(i, j, l) \in A$ denotes an arc from vertex $i$ to vertex $j$ labeled $l$.

In practice, we construct a complete graph, including parallel arcs with different labels, but excluding self-loops. Given arc weights $\boldsymbol{\mu} \in \mathbb{R}^A$, argument identification reduces to the selection of the subset of arcs of maximum weight such at most one arc with a given direction between any two vertices is selected. Again, this problem is simple. We denote a set of arcs as a boolean vector $\boldsymbol{z} \in \{0,1\}^A$ where $z_{i,j,l} = 1$ indicates that there is an arc from vertex $i$ to vertex $j$ labeled $l$ in the prediction.

**Multiple concepts per words.** More realistic semantic parsing scenarios require to trigger more than one concept per word. This only impacts the concept tagging step: one must change the model to allow the prediction of several concepts per word,

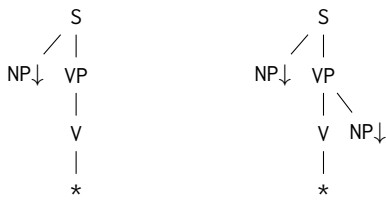

Figure 2: Two supertag examples from an LTAG. **(left)** Supertag associated with an intransitive verb. The substitution site NP↓ indicates the position of the subject. **(right)** Supertag associated with a transitive verb. The supplementary substitution site on the right indicates the position of the object of the verbal phrase.

for example via multi-label prediction or using several tagging layers (Jambor and Bahdanau, 2022). The argument identification step is left unchanged, *i.e.*, we create one vertex per predicted concept in the graph.

## 3 Supertagging for graph-based semantic parsing

In the syntactic parsing literature, supertagging refers to assigning complex descriptions of the syntactic structure directly at the lexical level (Bangalore and Joshi, 1999). For example, while an occurrence of the verb 'to walk' can be described in a coarse manner via its part-of-speech tag, a supertag additionally indicates that this verb appears in a clause with a subject on the left and a verbal phrase on the right, the latter also potentially requiring an object on its right, see Figure 2 for an illustration in the formalism of lexicalized Tree-Adjoining Grammars (LTAGs, Joshi et al., 1975).

We propose to introduce an intermediary *semantic supertagging* step in a graph-based semantic parser. The pipeline of Section 2.2 becomes:

1. concept tagging;

2. semantic supertagging;

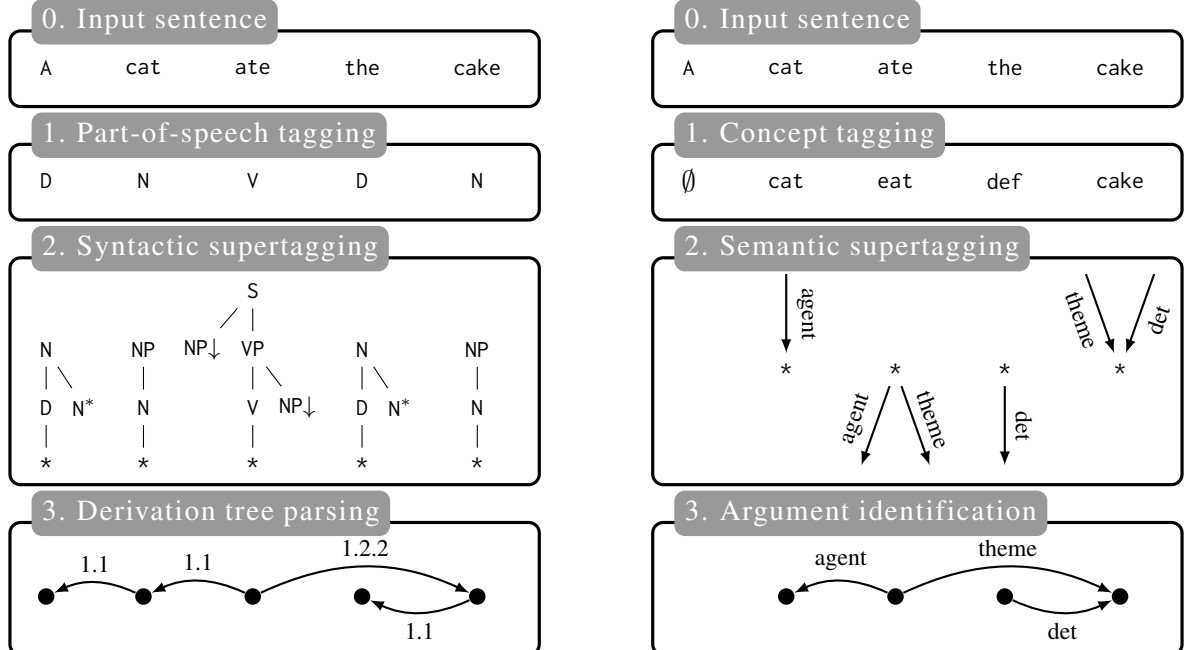

Figure 3: Comparaison of a LTAG parsing pipeline and the proposed semantic parsing pipeline. **(left)** A standard LTAG parsing pipeline. We start with an input sentence, then predict one part-of-speech tag per word. This part-of-speech can be used as feature or to filter the set of possible supertags in the next step. The supertagging step chooses one supertag per word. Finally, in the last step we merge all supertags together. This operation can be synthesised as a derivation tree where arc labels indicate Gorn adresses of substitution and adjunction operations. We refer readers unfamiliar with LTAG parsing to (Kallmeyer, 2010). **(right)** Our semantic parsing pipeline. We start with an input sentence, then predict at most one concept per word, where $\emptyset$ indicates no concept. The supertagging step assigns one supertag per concept instance (*i.e.*, excluding words tagged with $\emptyset$). Finally, the argument identification step identifies arguments of predicates using the valency constraints from the supertags.

3. argument identification.

This new pipeline is illustrated on Figure 3. Note that the introduction of the novel step does not impact the concept tagging step. As such, our approach is also applicable to datasets that would require multiple concepts prediction per word (see Section 2.2).

### 3.1 Semantic supertagging

In our setting, a supertag indicates the expected arguments of a concept instance (potentially none for an entity) and also how the concept is used. Contrary to syntactic grammars, our supertags do not impose a direction. In the following, we refer to an expected argument as a **substitution site** and to an expected usage as a **root**.

Formally, we define a (semantic) supertag as a multiset of tuples $(l, d) \in L \times \{-, +\}$ where $l$ is a label and $d$ indicates either substitution site or root, *e.g.*, $(\text{agent}, -)$ is a substitution site and $(\text{agent}, +)$ is a root. For example, in Figure 1, the supertag associated with 'like' is $[\![(\text{agent}, -), (\text{ccomp}, -)]\!]$

and the one associated with 'prefer' is $[\![(\text{agent}, -), (\text{xcomp}, -), (\text{ccomp}, +)]\!]$. The set of all supertags is denoted $S$.

**The Companionship principle (CP).**[2] Let us first consider the following simple example: assuming we would like to parse the sentence "Marie ate", yet associate the transitive supertag $[\![(\text{agent}, -), (\text{theme}, -)]\!]$ to the verb. In this case, the argument identification step will fail: the verb has no object in this sentence. The CP states that each substitution site must have a potential root in the supertag sequence. That is, in the supertagging step, we must make sure that the number of substitution sites with a given label exactly matches the number of roots with the same label, to ensure that there will exist at least one feasible solution for the next step of the pipeline. As such, supertagging here is assigning tags in context.

**Theorem 1.** *Given a set of supertags, a sequence of concept instances and associated supertag weights, the following problem is NP-complete: is there*

---

[2]We borrow the name from (Bonfante et al., 2009, 2014), although our usage is slightly different.

*a sequence of supertag assignments with linear weight $\geq m$ that satisfies the CP?*

*Proof.* First, note that given a sequence of supertags, it is trivial to check in linear time that its linear weight is $\geq m$ and that it satisfies the CP, therefore the problem is in NP. We now prove NP-completeness by reducing 3-dimensional matching to supertagging with the CP.

3-dim. matching is defined as follows: Let $A = \{a(i)\}_{i=1}^n$, $B = \{b(i)\}_{i=1}^n$ and $C = \{c(i)\}_{i=1}^n$ be 3 sets of $n$ elements and $D \subseteq A \times B \times C$. A subset $D' \subseteq D$ is a 3-dim. matching if and only if, for any two distinct triples $(a, b, c) \in D'$ and $(a', b', c') \in D'$, the following three conditions hold: $a \neq a'$, $b \neq b'$ and $c \neq c'$.

The following decision problem is known to be NP-complete (Karp, 1972): given A, B, C and D, is there a 3-dim. matching $D' \subseteq D$ with $|D'| \geq n$?

We reduce this problem to supertagging with the CP as follows. We construct an instance of the problem with $3n$ concept instances $a(1), ..., a(n), b(1), ..., b(n), c(1), ..., c(n)$. The supertag set $S$ is defined as follows, where their associated weight is 0 except if stated otherwise:

- For each triple $(a, b, c) \in D$, we add a supertag $[\![(b, -), (c, -)]\!]$ with weight 1 if and only if it is predicted for concept $a$;

- For each $b \in B$, we add a supertag $[\![(b, +)]\!]$ with weight 1 if and only if it is predicted for concept $b$;

- For each $c \in C$, we add a supertag $[\![(c, +)]\!]$ with weight 1 if and only if it is predicted for concept $c$.

If there exists a sequence of supertag assignment satisfying the CP that has a weight $\geq m = 3n$, then there exists a solution for the 3-dim. matching problem, given by the supertags associated with concept instances $a(1), ..., a(n)$. $\square$

Note that an algorithm for the supertagging decision problem could rely on the maximisation variant as a subroutine. This result motivates the use of a heuristic algorithm. We rely on the continuous relaxation of an integer linear program that we embed in a branch-and-bound procedure. We first explain how we construct the set of supertags as it impacts the whole program.

**Supertag extraction.** To improve generalization capabilities, we define the set of supertags as containing (1) the set of all observed supertags in the training set, augmented with (2) the cross-product of all root combinations and substitution site combinations. For example, if the training data contains supertags $[\![(\text{ccomp}, +), (\text{agent}, -)]\!]$ and $[\![(\text{agent}, -), (\text{theme}, -)]\!]$, we also include $[\![(\text{ccomp}, +), (\text{agent}, -), (\text{theme}, -)]\!]$ and $[\![(\text{agent}, -)]\!]$ in the set of supertags.

Formally, let $S^+$ (resp. $S^-$) be the set of root combinations (resp. substitution site combinations) observed in the data. The set of supertags is:

$$S = \left\{ s^+ \cup s^- \, \middle| \, \begin{array}{c} s^+ \in S^+ \, \wedge \, s^- \in S^- \\ \wedge \, s^+ \cup s^- \neq [\![\,]\!] \end{array} \right\}.$$

Note that the empty multiset can not be a supertag.

**Supertag prediction.** Let $\boldsymbol{y}^- \in \{0, 1\}^{n \times S^-}$ and $\boldsymbol{y}^+ \in \{0, 1\}^{n \times S^+}$ be indicator variables of the substitution sites and roots, respectively, associated with each word, *e.g.* $y_{i,s}^- = 1$ indicates that concept instance at position $i \in [n]$ has substitution sites $s \in S^-$. We now describe the constraints that $\boldsymbol{y}^-$ and $\boldsymbol{y}^+$ must satisfy. First, each position in the sentence should have exactly one set of substitution sites and one set of roots if and only if they have an associated concept:

$$\sum_{s \in S^-} y_{i,s}^- = 1 - x_{i,\emptyset} \qquad \forall i \in [n] \quad (1)$$

$$\sum_{s \in S^+} y_{i,s}^+ = 1 - x_{i,\emptyset} \qquad \forall i \in [n] \quad (2)$$

Next, we forbid the empty supertag:

$$y_{i,[\![\,]\!]}^- + y_{i,[\![\,]\!]}^+ \leq 1 \qquad \forall i \in [n] \quad (3)$$

Finally, we need to enforce the companionship principle. We count in $v_{s,l}^-$ the number of substitution sites with label $l \in L$ in $s \in S^-$, and similarly in $v_{s,l}^+$ for roots. We can then enforce the number of roots with a given label to be equal to the number of substitution sites with the same label as follows:

$$\sum_{\substack{i \in [n], \\ s \in S^-}} y_{i,s}^- v_{s,l}^- = \sum_{\substack{i \in [n], \\ s \in S^+}} y_{i,j}^+ v_{s,l}^+ \qquad \forall l \in L. \quad (4)$$

All in all, supertagging with the companionship principle reduces to the following integer linear program:

$$\max_{\boldsymbol{y}^-, \boldsymbol{y}^+} \quad \langle \boldsymbol{y}^-, \boldsymbol{\phi}^- \rangle + \langle \boldsymbol{y}^+, \boldsymbol{\phi}^+ \rangle,$$

$$\text{s.t.} \quad (1\text{–}4),$$

$$\boldsymbol{y}^- \in \{0, 1\}^{n \times S^-}, \boldsymbol{y}^+ \in \{0, 1\}^{n \times S^+}.$$

In practice, we use the CPLEX solver.[3]

**Timing.** We initially implemented this ILP using the CPLEX Python API. The resulting implementation could predict supertags for only $\approx 10$ sentences per second. We reimplemented the ILP using the CPLEX C++ API (via Cython) with a few extra optimizations, leading to an implementation that could solve $\approx 1000$ instances per second.

## 3.2 Argument identification

The last step of the pipeline is argument identification. Note that in many cases, there is no ambiguity, see the example in Figure 3: as there is at most one root and substitution site per label, we can infer that the theme of concept instance eat is cake, etc. However, in the general case, there may be several roots and substitution sites with the same label. In the example of Figure 1, we would have 3 agent roots after the supertagging step.

For ambiguous labels after the supertagging step, we can rely on a bipartite matching (or assignment) algorithm. Let $l \in L$ be an ambiguous label. We construct a bipartite undirected graph as follows:

- The first node set $C$ contains one node per substitution site with label $l$;

- The second node set $C'$ contains one node per root with label $l$;

- we add an edge for each pair $(c, c') \in C \times C'$ with weight $\mu_{i,j,l}$, where $i \in [n]$ and $j \in [n]$ are sentence positions of the substitution site represented by $c$ and the root represented by $c'$, respectively.

We then use the Jonker-Volgenant algorithm (Jonker and Volgenant, 1988; Crouse, 2016) to compute the matching of maximum linear weight with complexity cubic w.r.t. the number of nodes. Note that thanks to the companionship principle, there is always at least one feasible solution to this problem, *i.e.*, our approach will never lead to a "dead-end" and will always predict a (potentially wrong) semantic parse for any given input.

## 4 Training objective

**Supervised loss.** Let $(\widehat{x}, \widehat{y}^-, \widehat{y}^+, \widehat{z})$ be a gold annotation from the training dataset. We use separable negative log-likelihood losses (NLL) for each step as they are fast to compute and work well in

[3] https://www.ibm.com/products/ilog-cplex-optimization-studio

practice (Zhang et al., 2017; Corro, 2023). The concept loss is a sum of one NLL loss per word:

$$\ell_{\text{concept}}(\boldsymbol{\lambda}; \widehat{\boldsymbol{x}}) = -\langle \boldsymbol{\lambda}, \widehat{\boldsymbol{x}} \rangle + \sum_{i \in [n]} \log \sum_{t \in T} \exp \lambda_{i,t}.$$

For supertagging, we use the following losses:

$$\ell_{\text{sub.}}(\boldsymbol{\phi}^-; \widehat{\boldsymbol{y}}^-) = -\langle \boldsymbol{\phi}^-, \widehat{\boldsymbol{y}}^- \rangle$$
$$+ \sum_{i \in [n]} \log \sum_{s \in S^-} \exp \phi_{i,s}^-,$$
$$\ell_{\text{root}}(\boldsymbol{\phi}^+; \widehat{\boldsymbol{y}}^+) = -\langle \boldsymbol{\phi}^+, \widehat{\boldsymbol{y}}^+ \rangle$$
$$+ \sum_{i \in [n]} \log \sum_{s \in S^+} \exp \phi_{i,s}^+.$$

Finally, for argument identification we have one loss per couple of positions in the sentence:

$$\ell_{\text{arg.}}(\boldsymbol{\mu}; \boldsymbol{z}) = -\langle \boldsymbol{\mu}, \boldsymbol{z} \rangle$$
$$+ \sum_{(i,j) \in [n] \times [n]} \log \sum_{l \in L} \exp \mu_{i,j,l}.$$

Note that for the concept loss, we have a special empty tag with null score for the case where there is no concept associated with a word in the gold output (and similarly for argument identification).

**Weakly-supervised loss.** In practice, it is often the case that we do not observe the alignment between concept instances and words in the training dataset, which must therefore be learned jointly with the parameters. To this end, we follow an "hard" EM-like procedure (Neal and Hinton, 1998):

- E step: compute the best possible alignment between concept instances and words;

- M step: apply one gradient descent step using the "gold" tuple $(\widehat{x}, \widehat{y}^-, \widehat{y}^+, \widehat{z})$ induced by the alignment from the E step.

Note that the alignment procedure in the E step is NP-hard (Petit and Corro, 2023, Theorem 2), as the scoring function is not linear. For example, assume two concept instances $p$ and $p'$ such that $p'$ is an argument of $p$. If $p$ and $p'$ are aligned with $i$ and $j$, respectively, the alignment score includes the token tagging weights induced by this alignment plus the weight of the labeled dependency from $i$ to $j$.

We propose to reduce the E step to *maximum a posteriori* (MAP) inference in a factor graph, see Figure 4. We define one random variable (RV) taking values in $[n]$ per concept instance. The assignment of these RVs indicate the alignment between

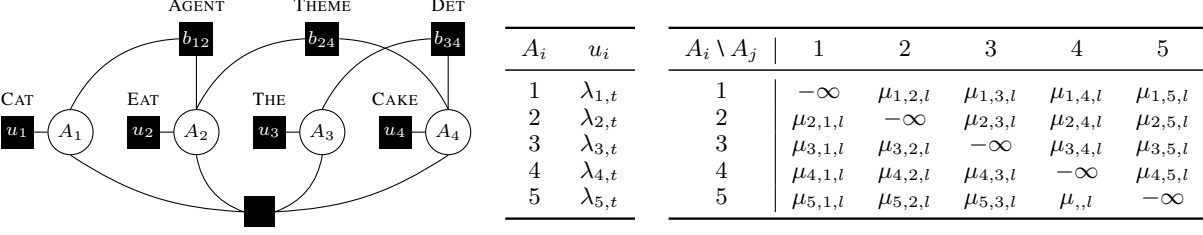

Figure 4: **(left)** The factor graph used to compute the best alignment of the semantic graph in E steps for the sentence 'A cat ate the cake'. Each variable corresponds to a concept instance in the semantic graph and has an associated unary factor. For each argument of a concept instance, we add a binary factor between between the concept instance and its argument's instance. **(center)** Weights of an unary factor given the value of its corresponding variable. We denote $t$ the concept corresponding to that factor (displayed above unary factors). **(right)** Weights of a binary factor $b_{ij}$, where the concept instance represented by $A_j$ is an argument of the concept instance represented by $A_i$ in the semantic graph. We denote $l$ the label of the argument (displayed above binary factors).

concept instances and words. Unary factors correspond to tagging weights, *e.g.* aligning a concept $t \in T$ with word $i \in [n]$ induces weight $\lambda_{i,t}$. Binary factors correspond to argument identification: for each arc the semantic graph, we add a binary factor between the two concept instances RVs that will induce the dependency weight given the RVs assignment. Finally, there is a global factor acting as an indicator function, that forbids RVs assignments where different concept instances are aligned with the same word. We use AD3 (Martins et al., 2011) for MAP inference in this factor graph.

## 5 Related work

**Compositional generalization.** Compositional generalization has been a recent topic of interest in semantic parsing. This is because failure to generalize is an important source of error, especially in seq2seq models (Lake and Baroni, 2018; Finegan-Dollak et al., 2018; Herzig and Berant, 2021; Keysers et al., 2020). Several directions have been explored in response. Zheng and Lapata (2021) rely on latent concept tagging in the encoder of a seq2seq model, while Lindemann et al. (2023) introduce latent fertility and re-ordering layers in their model. Another research direction uses data augmentation methods to improve generalization (Jia and Liang, 2016; Andreas, 2020; Akyürek et al., 2021; Qiu et al., 2022; Yang et al., 2022).

Span-based methods have also been shown to improve compositional generalization (Pasupat et al., 2019; Herzig and Berant, 2021; Liu et al., 2021). Particularly, Liu et al. (2021) explicitly represent input sentences as trees and use a Tree-LSTM (Tai et al., 2015) in their encoder. While this

parser exhibits strong performance, this approach requires work from domain experts to define the set of operations needed to construct trees for each dataset. Other line of work that seek to tackle compositional generalization issues include using pretrained models (Herzig et al., 2021; Furrer et al., 2021), specialized architectures (Korrel et al., 2019; Russin et al., 2020; Gordon et al., 2020; Csordás et al., 2021) and regularization (Yin et al., 2023).

**Graph-based semantic parsing.** Graph-based methods have been popularized by syntactic dependency parsing (McDonald et al., 2005). To reduce computational complexity, Dozat and Manning (2018) proposed a neural graph-based parser that handles each dependency as an independent classification problem. Similar approaches were applied in semantic parsing, first for AMR parsing (Lyu and Titov, 2018; Groschwitz et al., 2018). Graph-based approaches have only recently been evaluated for compositional generalization. The approach proposed by Petit and Corro (2023) showed significant improvements compared to existing work on compositional splits of the GeoQuery dataset. However, their parser can only generate trees. Weißenhorn et al. (2022) and Jambor and Bahdanau (2022) introduced approaches that can handle arbitrary graphs, a requirement to successfully parse COGS.

## 6 Experiments

We use a neural network based on a BiLSTM (Hochreiter and Schmidhuber, 1997) and a biaffine layer for arc weights (Dozat and Manning, 2017). More detail are given in Appendix A. As usual in the compositional generalization literature, we

| | Structural gen. | | | Lexical gen. | Overall |
|---|---|---|---|---|---|
| | Obj to Subj PP | PP recursion | CP recursion | | |
| **Seq2seq models** | | | | | |
| Kim and Linzen (2020) | 0 | 0 | 0 | 42 | 35 |
| Conklin et al. (2021)† | - | - | - | - | 67 |
| Akyürek et al. (2021) | 0 | 1 | 0 | 96 | 83 |
| Zheng and Lapata (2021) | 0 | 39 | 12 | 99 | 89 |
| **Structured models** | | | | | |
| LEAR (Liu et al., 2021) | - | - | - | - | 97.7 |
|   w/o Tree-LSTM | - | - | - | - | 80.7 |
|   reproduction by Weißenhorn et al. (2022) | **93** | 99 | **100** | 99 | **99** |
| Jambor and Bahdanau (2022)† | - | - | - | - | 82.3 |
| Weißenhorn et al. (2022) | 59 | 36 | **100** | 82 | 79.6 |
| **Our baselines: Standard graph-based parser** | | | | | |
| Full model | 11.6 | 0 | 0 | 97.4 | 84.1 |
|   w/o early stopping | 12.7 | 0 | 0 | 97.3 | 84.1 |
|   w/o early stopping & w/o supertagging loss | 9.8 | 0 | 0 | 97.5 | 84.1 |
| **Proposed method: graph-based parser with supertagging** | | | | | |
| Full model | 75.0 | **100** | **100** | **99.1** | 98.1 |
|   w/o early stopping | 51.1 | **100** | **100** | 98.9 | 96.7 |

Table 2: Exact match accuray on COGS. We report results for each subset of the test set (structural generalization and lexical generalization) and the overall accuracy. For our results, we report the mean over 3 runs. Entries marked with † use a subset of 1k sentences from the generalization set as their development set.

evaluate our approach in a fully supervised setting, *i.e.*, we do not use a pre-trained neural network like BERT (Devlin et al., 2019). Code to reproduce the experiments is available online.[4]

## 6.1 Early stopping

COGS only possesses an in-distribution development set and the accuracy of most parsers on this set usually reaches 100%. Previous work by Conklin et al. (2021) emphasized that the lack of a development set representative of the generalization set makes model selection difficult and hard to reproduce. They proposed to sample a small subset of the generalization set that is used for development. Both their work and LaGR (Jambor and Bahdanau, 2022) use this approach and sample a subset of 1000 sentences from the generalization set to use as their development set. However, we argue that this development set leaks compositional generalization information during training.

We propose a variant of early stopping to prevent overfitting on the in-distribution data without requiring a compositional generalization development set. We incrementally freeze layers in the neural network as follows: each subtask (predic-

tion of tags, supertags, dependencies) is monitored independently on the in-distribution development set. As soon as one of these tasks achieves 100% accuracy, we freeze the shared part of the neural architecture (word embeddings and the BiLSTM). We also freeze the layers that produce the scores of the perfectly predicted task. For each subsequent task that achieves perfect accuracy, the corresponding layers are also frozen. This early stopping approach prevents overfitting.

We also experimented using the hinge loss instead of the NLL loss as it shares similar properties to our early stopping strategy: once a prediction is correct (including a margin between the gold output and other outputs), the gradient of the loss becomes null. We however found that this loss yields very low experimental results (null exact match score on the test set).

## 6.2 Results

All results are exact match accuracy, *i.e.*, the ratio of semantic structures that are correctly predicted. We report the overall accuracy,[5] the accuracy over all lexical generalization cases as well as the individual accuracy for each structural generalization

[4]https://github.com/alban-petit/semantic-supertag-parser

[5]As COGS contains 1,000 sentences for each generalization, case, this number mostly reflects the accuracy for lexical generalization, which account for 85.7% of the test set.

| | Obj to Subj PP | PP rec. | CP rec. |
|---|---|---|---|
| **Word level accuracy** | | | |
| ILP | 90.2 | 100 | 100 |
| No ILP | 71.6 | 99.9 | 100 |
| **Sentence level accuracy** | | | |
| ILP | 75.0 | 100 | 100 |
| No ILP | 9.0 | 99.6 | 100 |

Table 3: Supertagging accuracy using our integer linear program (ILP) and without (*i.e.* simply predicting the best supertag for each word, without enforcing the companionship principle).

case. We report mean accuracy over 3 runs.

**External baselines.** We compare our method to several baselines: (1) the seq2seq models of Kim and Linzen (2020), Akyürek et al. (2021) and Zheng and Lapata (2021); (2) two graph-based models, LAGR (Jambor and Bahdanau, 2022) and the AM parser of Weißenhorn et al. (2022); (3) LeAR (Liu et al., 2021), a semantic parser that relies on a more complex Tree-LSTM encoder (Tai et al., 2015). We also report the performance of LeAR when a BiLSTM is used in the encoder instead of the Tree-LSTM.

**Our baselines.** We also report results for our model using the standard graph-based semantic parsing pipeline (Section 2.2), that is without the intermediary supertagging step. Note that, in this case, the supertagging loss becomes an auxiliary loss, as proposed by Candito (2022).

**Result comparison.** We observe that our approach outperforms every baseline except LEAR. Importantly, our method achieves high exact match accuracy on the structural generalization examples, although the *Obj to subj PP* generalization remains difficult (our approach only reaches an accuracy of 75.0% for this case).

We now consider the effect of our novel inference procedure compared to our standard graph-based pipeline. It predicts *PP recursion* and *CP recursion* generalizations perfectly, where the baseline accuracy for these cases is 0. For *Obj to subj PP* generalization, our best configuration reaches an accuracy of 75.0%, 5 times more than the baselines. All in all, the proposed inference strategy improves results in the three structural generalizations subsets, and brings lexical generalization cases closer to 100% accuracy.

**Impact of training procedure.** The early stopping approach introduced above has a clear impact

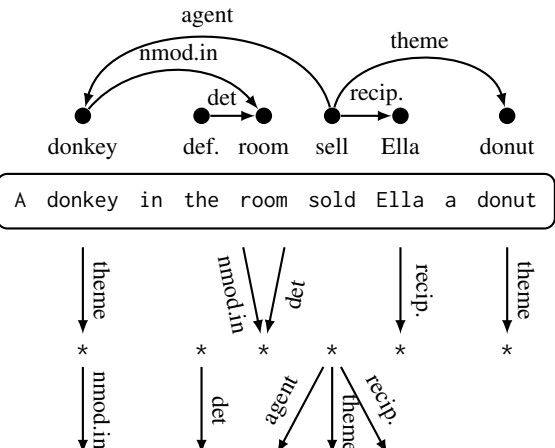

Figure 5: **(top)** Gold semantic graph. **(bottom)** Supertags predicted without enforcing the companionship principle. A mistake occurs for 'donkey' as the theme root is predicted, instead of agent. This is probably due to the introduction of a PP before the verb, which confuses the network: PP only occur with objects during training. Using ILP fixes this mistake.

for *Obj to subj PP*, resulting in a 23.9 points increase (from 51.1 to 75.0). Such improvements are not observed for the baselines. From this, we conclude that our neural architecture tends to overfit the COGS training set and that some measures must be taken to mitigate this behaviour.

**Suppertagging accuracy.** We report in Table 3 the supertagging accuracy with and without enforcing the companionship principle. We observe a sharp drop in accuracy for the *Obj to Subj PP* generalization when the companionship principle is not enforced. This highlights the importance of structural constraints to improve compositional generalization. We observe that the many error are due to the presence of the prepositional phrase just after the subject: this configuration causes the supertagger to wrongly assign a theme root to the subject, instead of agent. When the companionship principle is enforced, this mistake is corrected. An illustration is in Figure 5.

## 7 Conclusion

We proposed to introduce a supertagging step in a graph-based semantic parser. We analysed complexities and proposed algorithms for each step of our novel pipeline. Experimentally, our method significantly improves results for cases where compositional generalization is needed.

## Limitations

One limitation of our method is that we cannot predict supertags unseen during training (*e.g.*, combinaison of roots unseen at training time). Note however that this problem is well-known in the syntactic parsing literature, and meta-grammars could be used to overcome this limitation. Another downside of our parser is the use of an ILP solver. Although it is fast when using the COGS dataset, this may be an issue in a more realistic setting. Finally, note that our method uses a pipeline, local predictions in the first steps cannot benefit from argument identification scores to fix potential errors.

## Acknowledgments

We thank the anonymous reviewers and meta-reviewer for their comments and suggestions. This work was funded by the UDOPIA doctoral program in Artifial Intelligence from Université Paris-Saclay (ANR-20-THIA-0013) and benefited from computations done on the Saclay-IA platform.

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

## A Neural architecture

The neural architecture used in our experiments to produce the weights $\boldsymbol{\lambda}$, $\phi^+$, $\phi^-$ and $\boldsymbol{\mu}$ is composed of:

- An embedding layer of dimension 200 followed by a bi-LSTM (Hochreiter and Schmidhuber, 1997) with a hidden size of 400.

- A linear projection of dimension 300 followed by a RELU activation and another linear projection of dimension $|T|$ to produce $\boldsymbol{\lambda}$.

- A linear projection of dimension 200 followed by a RELU activation and another linear projection of dimension $|S^+|$ to produce $\phi^+$.

- A linear projection of dimension 200 followed by a RELU activation and another linear projection of dimension $|S^-|$ to produce $\phi^-$.

- A linear projection of dimension 200 followed by a RELU activation and a bi-affine layer to produce $\boldsymbol{\mu}$.

We apply dropout with a probability of 0.3 over the outputs of each layer except the final layer for each weight matrix. The learning rate is $5 \times 10^{-4}$ and there are 30 sentences per mini-batch.