# OpenReview forum: "Structural generalization in COGS: Supertagging is (almost) all you need"
_EMNLP/2023/Conference — EMNLP 2023 Main_

### Official Review · Reviewer_6hiA · 2023-08-01

**Typos Grammar Style And Presentation Improvements:** 1. line 250
**Soundness:** 4

**Excitement:**

4: Strong: This paper deepens the understanding of some phenomenon or lowers the barriers to an existing research direction.

**Paper Topic And Main Contributions:**

This paper focuses on generalizations from substructures in the field of parsing for semantic graph representations. The authors define supertags that contain local graph fragments (incoming and outgoing edges with label) and add their prediction as a prepocessing step in the parsing process. The predicted supertags are constrained such that they must form a valid derivation. Their experiments suggest that this addition to the parsing pipeline increases overall parsing accuracy and specific scores intended to measure structural generalization in COGS compared to a baseline without the supertagging stage.

**Questions For The Authors:**

1. The restrictions imposed by equation 4 in ln 268 does not take loops into account, right? Is there some other constraint that would prevent a combination of supertags that could only form a graph with a loop or did I miss something else entirely?
2. Can you explain the reasoning behind the weakly supervised loss? I don't understand why there is a concept tagging stage if these are not part of the corpus anyways. At some point I convinced myself these are used to be able to assign an empty concept tag that filters "unproductive" words for the remainder of the parsing pipeline, but that would also be possible and easier to formulate with an empty supertag, imho.

**Reasons To Accept:**

To me, their additions seems like a solid approach to formalize local substructures in graphs and their experimental results give the same impression. The COGS corpus enables a focused evaluation of the structural generalization capabilities and fits very well to the contribution.

**Reasons To Reject:**

There are some minor inconsistencies in notation and some details about the approach that I (as someone unfamiliar with this specific topic) did not completely understand (listed below). However, these could be easily ironed out in a camera-ready version.

**Reproducibility:**

4: Could mostly reproduce the results, but there may be some variation because of sample variance or minor variations in their interpretation of the protocol or method.

**Reviewer Confidence:**

2: Willing to defend my evaluation, but it is fairly likely that I missed some details, didn't understand some central points, or can't be sure about the novelty of the work.

---

> ### Author Rebuttal · Authors · 2023-08-24
>
> Dear reviewer,
>
> Thank you for your review.
>
> Indeed, there is a mistake line 250. A supertag is a multiset, and S is a set of supertags. The empty supertag is an empty multiset $[[ ]]$. Hence, the correct formula is what you proposed (slighthly modified so that the empty supertag is not allowed):
> $$
> S = \\{ s^- \cup s^+ | s^- \in S^-, s^+ \in S^+, s^- \cup s^+ \neq [[ ]] \\}
> $$
>
> "The restrictions imposed by equation 4 in ln 268 does not take loops into account, right? "
>
> No, it does not prevent self-loops, but they never appear in practice.
>
> "Can you explain the reasoning behind the weakly supervised loss?"
>
> There is a concept tagging stage (that also identifies semantically empty words --- i.e. words that do not trigger any concept). However, in many semantic parsing datasets, we do not know to which words are concept anchors, i.e. we cannot learn the concept tagger in a supervised fashion. For a given sentence all we observe is a sequence of words and a list of concepts, with no explicit correspondance between the two. The weakly supervised loss learns this alignment between words and concepts in an unsupervised fashion.

---

### Official Review · Reviewer_KQiC · 2023-08-01

**Typos Grammar Style And Presentation Improvements:** Line 484
**Soundness:** 4

**Excitement:**

4: Strong: This paper deepens the understanding of some phenomenon or lowers the barriers to an existing research direction.

**Paper Topic And Main Contributions:**

Inspired by traditional NLP supertagging success, the paper investigates in applying similar techniques to compositional semantic parsing. More concretely, between the more common graph parsing steps where it involves concept tagging and argument identification, the proposed algorithm inserts a semantic supertagging step between the above two steps.

The paper also contains some technical novelties put forward including that the argument identification is reduced to graph matching and the paper proposes a novel early stopping criteria with dev distribution different from the test distribution.

The paper tests the proposed framework on COGs and show that the supertagging introduction significantly improves the performance and the introduced techniques also help the performance.

**Reasons To Accept:**

Supertagging techniques and related techniques such as CCG are an important part of NLP literature but it is difficult to find its position for NLP today. This paper shows in a narrow semantic parsing domain how those techniques can be used and combined with existing mainstream learning techniques.

The paper proposes some concrete technical contribution (e.g. early stopping based solely on distribution iid to training) that might generalize beyond the scope of this paper; the novelty has proved to have a practical performance impact on the tested dataset.

The authors discussed the limitations with transparency and insights.

**Reasons To Reject:**

Although it is understandable that the paper doesn't compare to pretrained models, it would still be appreciated if some literature can be covered for their performance on COGs. That would help to complete the picture for performance on this dataset.

Similarly, the authors only use LSTM and mainly LSTM baselines, it would be good to know the transformer performance on this dataset as well (e.g. https://arxiv.org/pdf/2108.12284.pdf), for the reason of completeness.

**Reproducibility:**

3: Could reproduce the results with some difficulty. The settings of parameters are underspecified or subjectively determined; the training/evaluation data are not widely available.

**Reviewer Confidence:**

3: Pretty sure, but there's a chance I missed something. Although I have a good feel for this area in general, I did not carefully check the paper's details, e.g., the math, experimental design, or novelty.

---

> ### Author Rebuttal · Authors · 2023-08-24
>
> Dear reviewer,
>
> Thank you for your review.
>
> Indeed, we can extend the related work section to include analyses conducted with pre-trained neural networks. We apologize for this unintentional omission.

---

### Official Review · Reviewer_Gx4y · 2023-08-05

**Soundness:** 3

**Excitement:**

2: Mediocre: This paper makes marginal contributions (vs non-contemporaneous work), so I would rather not see it in the conference.

**Paper Topic And Main Contributions:**

The papers extends a graph-based neural semantic parser with an additional intermediate supertagging step, the results of which are used for subsequent argument identification for the semantic parser. The supertagging accuracies are in the range of 75 to 100 on 3 different types, and when plugged into the semantic parser it helps significantly over the baseline without supertagging. The supertagging is realised using ILP. Compared with fully sequential baseline models such as seq2seq models the proposed model outperforms these; however, in comparison with structural models such as Tree-LSTMs it's on par on most categories except 1.

**Reasons To Accept:**

* Semantic parsing still has some relevance in some domains
* Supertagging used to reduce the search space of a semantic parser which shows some positive results

**Reasons To Reject:**

* The model is only competitive with the baseline graph-based parser but not to other structural models such as Tree-LSTMs which is more flexible than a supertagging and would be more easily adapted to new domains
* Supertagging as an intermediate step may introduce more errors into the 3-stage semantic parsing pipeline in the paper which could be alleviated at least partially with an end2end model
* Supertagging then parsing introduces additional challenges in itself; the authors have shown a few theorems to prove otherwise, but empirically it's widely known the large amount of ambiguity present after supertagging may be hard to resolve and in combination with parsing, it may make it even harder and this challenge itself may require neural models, so why not just use an end2end neural model for semantic parsing?


**Reproducibility:**

3: Could reproduce the results with some difficulty. The settings of parameters are underspecified or subjectively determined; the training/evaluation data are not widely available.

**Reviewer Confidence:**

4: Quite sure. I tried to check the important points carefully. It's unlikely, though conceivable, that I missed something that should affect my ratings.

---

> ### Author Rebuttal · Authors · 2023-08-24
>
> Dear reviewer,
>
> Thank you for your review. We try to address your main concerns below.
>
> First, we want to emphasize that the term **supertagging** is used as an analogy to supertagging in the syntactic parsing litterature, but that our approach is simpler and does not suffer from many drawbacks of LTAG and CCG supertagging:
> - our semantic supertags are trivial to extract from the training corpus, and we do not need any heuristic to extract them as required for LTAGs (head-percolation tables, rules to decide if an attachment should be made via substitution or adjunction, etc);
> - in syntactic parsing, supertagging lowers the grammar size impact in the parsing step but may lead to an empty parse forest --- this is not the case of our approach thanks to the companionship principle constraint;
> - in syntactic parsing, the parsing step may still have a high complexity (e.g. O(n^7) in the case of LTAGs) --- this is not the case of our approach where the last step admits a fast cubic-time algorithm by reduction to bipartite matching for which many efficient implementations exist.
>
> ---
>
> *"[...] such as Tree-LSTMs which is more flexible than a supertagging and would be more easily adapted to new domains"*
>
> We beg to differ here:
>
> (1) There is nothing to do in order to adapt our supertagger to a new domain. All supertags are automatically extracted from training data (lines 237-246).
>
> (2) On the contrary, the tree-LSTM based approach of Liu et al. (2021), called LEAR, requires domain expert adaptation of the composition operation in their model, see Section 6.3 of their paper: "In addition, while our approach is general for various semantic parsing tasks, the collection of semantic operations needs to be redesigned for each task. We need to ensure that these semantic operations are k-ary projections [..], and all the meaning representations are covered by the operations collection. This is tractable, but still requires some efforts from domain experts"
>
> As such, our approach is easier than Liu et al's to adapt to new domains (in fact it is straightforward --- there is nothing to do).
>
> ---
>
> *"The model is only competitive with the baseline graph-based parser but not to other structural models such as Tree-LSTMs"*
>
> Our approach achieves better or comparable scores compared to all previous models **except** on one type of generalization (Obj to Subj PP) where a single model (Tree-LSTM LEAR) achieves better results. Moreover, as stated above, Tree-LSTM LEAR requires very specific domain related configuration. Finally, Tree-LSTM LEAR requires many extra tricks to work: for example, in Section 4.2 of Liu et al (2021), they explain that they need to introduce constraints in the latent structure inference procedure and that their model cannot predict concepts spanning unseen word sequences (cf. 'Phrase Table Constraint' paragraph). Our model has no such limitation.
>
> ---
>
> *"Supertagging as an intermediate step may introduce more errors into the 3-stage semantic parsing pipeline in the paper which could be alleviated at least partially with an end2end model"*
>
> There are two types of errors that can be introduced by intermediary steps in a pipeline:
>
> (1) errors that makes the following steps infeasible: we ensure our approach never yield this kind of error via the "companionship principle"
>
> (2) prediction errors: it is known that end-to-end models fails on compositional generalization problems. On the contrary, our pipeline achieves better results.
>
> Moreover, note that many methods improving compositional generalization are based on pipelines. For example, the Tree-LSTM LEAR approach of Liu et al (2021) is based on a pipeline method (see Section 3 of their paper where they have first a composer and then an interpreter, and Figure 5 where they show a failure example due of the first step of their pipeline).
>
> ---
>
> *"Supertagging then parsing introduces additional challenges in itself; the authors have shown a few theorems to prove otherwise, but empirically it's widely known the large amount of ambiguity present after supertagging may be hard to resolve and in combination with parsing, it may make it even harder and this challenge itself may require neural models, so why not just use an end2end neural model for semantic parsing?"*
>
> We show in the paper that resolving can be reduced to bipartite matching (or assignment) in Section 3.2. Moreover, constraints in our supertagging scheme ensure  that the last step is feasible.
>
> On *"this challenge itself may require neural models"*: note that a standard graph based parser has, on top of a neural encoder, two distinct layers that produce nodes scores and arc scores. To this, our method only adds one extra layer for supertag scores.
>
>
> Finally, we experimentally demonstrate that a simpler approach, without a supertagger, fails on COGS (which was known and previous papers have similar results), see "Our baselines: Standard graph-based parser" in Table 2.
>
> ---
>
> *"3: Could reproduce the results with some difficulty. The settings of parameters are underspecified or subjectively determined; the training/evaluation data are not widely available."*
>
> The COGS dataset is available freely on github.
>
> Our neural network is very simple and fully described in appendix B, including training hyperparameters.

---

### Meta-Review · Area_Chair_K5J3 · 2023-09-19

**Recommendation:** 5

**Metareview:**

We had really abundant discussions regarding this work, which help clarify many important technique issues. Thank  reviewer Gx4y for the patience and hard work. Also thank the authors for detailed responses. Reviewer 6hiA resolved several math notation problems.
Based on the comments of all reviewers and authors’ rebuttal and my own reading of this paper, I think this work proposes a novel and correct three-step semantic parsing approach, which works well on the COGS dataset, especially from the structural generalization perspective.

Three minor suggestions for improving this paper:

1)	For the first step, please give more explicit discussions on the one-word-one-concept assumption. For example, how does the assumption affect the proposed approach in the sense that how or whether it can be applied to other datasets, especially those in which the number of concepts are often larger than that of words.

2)	I have a feeling that more results and analysis can make this more influential, especially from the  structural generalization perspective. Please use the extra page properly.

3)	Can you add discussion about the following recent work (ACL-2023) in the next version? I think there is strong connection between your work and this one. Of course I understand the two approaches are different.

> Compositional Generalization without Trees using Multiset Tagging and Latent Permutations Matthias Lindemann | Alexander Koller | Ivan Titov

---

### Decision · Program_Chairs · 2023-10-07

**Decision:**

Accept-Main

**Comment:**

We had really abundant discussions regarding this work, which help clarify many important technique issues. Thank  reviewer Gx4y for the patience and hard work. Also thank the authors for detailed responses. Reviewer 6hiA resolved several math notation problems.
Based on the comments of all reviewers and authors’ rebuttal and my own reading of this paper, I think this work proposes a novel and correct three-step semantic parsing approach, which works well on the COGS dataset, especially from the structural generalization perspective.

Three minor suggestions for improving this paper:

1)	For the first step, please give more explicit discussions on the one-word-one-concept assumption. For example, how does the assumption affect the proposed approach in the sense that how or whether it can be applied to other datasets, especially those in which the number of concepts are often larger than that of words.

2)	I have a feeling that more results and analysis can make this more influential, especially from the  structural generalization perspective. Please use the extra page properly.

3)	Can you add discussion about the following recent work (ACL-2023) in the next version? I think there is strong connection between your work and this one. Of course I understand the two approaches are different.

> Compositional Generalization without Trees using Multiset Tagging and Latent Permutations Matthias Lindemann | Alexander Koller | Ivan Titov